# RainyScape: Unsupervised Rainy Scene Reconstruction using Decoupled Neural Rendering

## ABSTRACT

We propose RainyScape, an unsupervised framework for reconstructing clean scenes from a collection of multi-view rainy images. RainyScape consists of two main modules: a neural rendering module and a rain-prediction module that incorporates a predictor network and a learnable latent embedding that captures the rain characteristics of the scene. Specifically, based on the spectral bias property of neural networks, we first optimize the neural rendering pipeline to obtain a low-frequency scene representation. Subsequently, we jointly optimize the two modules, driven by the proposed adaptive direction-sensitive gradient-based reconstruction loss, which encourages the network to distinguish between scene details and rain streaks, facilitating the propagation of gradients to the relevant components. Extensive experiments on both the classic neural radiance field and the recently proposed 3D Gaussian splatting demonstrate the superiority of our method in effectively eliminating rain streaks and rendering clean images, achieving state-of-the-art performance. The constructed high-quality dataset and source code will be publicly available.

## CCS CONCEPTS

• **Computing methodologies** → **Reconstruction**; *3D imaging*; Computational photography.

## KEYWORDS

Rainy scene reconstruction, Neural rendering, Unsupervised loss

## 1 INTRODUCTION

Neural Radiance Field (NeRF) [29] has emerged as a groundbreaking technique for novel view synthesis by learning a continuous, volumetric representation of the scene through differentiable volume rendering. NeRF's ability to create highly realistic and consistent novel views with fine details has led to its widespread adoption in various applications, such as 3D reconstruction [40], surface reconstruction [3], 3D object editing [36], and large-scale scene reconstruction [53].

However, when the input images are degraded by various factors such as blur, noise, or rain, the rendering results inevitably exhibit obvious artifacts. To address this issue, recent works have proposed a range of task-specific solutions. For instance, Ma et al. [27] achieved clear scene reconstruction from images affected by camera

*ACM MM, 2024, Melbourne, Australia*

© 2024 Copyright held by the owner/author(s). Publication rights licensed to ACM.
ACM ISBN 978-x-xxxx-xxxx-x/YY/MM
https://doi.org/10.1145/nnnnnnn.nnnnnnn

motion blur or defocus blur by learning ray fusion within the blur kernel. Huang et al. [16] tackled the problem of input images with different dynamic ranges by adjusting the ray intensity using a camera response function and obtaining the final color with a learnable tone mapper. Chen et al. [6] extended the atmospheric scattering model into the volume rendering process to fit hazy images using two neural rendering processes. In contrast to these methods, our approach specifically targets the rainy scene reconstruction task, which can be seamlessly integrated with our observation of the neural rendering prior, as illustrated in Fig. 2. Moreover, the sparse and intermittent nature of rain precipitation in 3D space makes it challenging to represent through an additional neural rendering field. Furthermore, our proposed framework can be readily adapted to work with various rendering techniques, demonstrating its versatility and flexibility.

In this paper, we propose RainyScape, a decoupled neural rendering framework that is capable of reconstructing a rain-free scene from rainy images in an unsupervised fashion. First, we obtain a low-frequency representation of the scene through a selected radiance field rendering process (e.g., NeRF [29] or 3D Gaussian Splatting [19]), where the remaining high-frequency scene details and rain information are coupled together. We then characterize the rain in the scene using learnable rain embeddings composed of rainy scene state vectors, viewpoint state vectors, and camera parameters, and predict rain maps through a multilayer perceptron (MLP) and a CNN predictor. To decouple the high-frequency scene details from the rain streak, we propose an adaptive angle estimate strategy that significantly improves the distinguishability of rain by considering the directions along and perpendicular to the rain streak. We use the obtained angle information to design a gradient rotation loss. Combining our proposed network framework and unsupervised loss, we employ an alternating optimization strategy to update network parameters and rain embeddings. Additionally, to address the lack of multi-view rainy scene datasets, we render 10 sets of scenes using Maya [1], resulting in more consistent and realistic rain trails compared to data simulated by simple methods.

In summary, the main contributions of this paper lie in:

- we explore priors in neural rendering processes and propose a general rainy scene reconstruction framework;
- we represent the rain in the scene using rain embeddings and use a predictor to predict rain streaks;
- we introduce an adaptive scene rain streak angle estimate strategy and a corresponding gradient rotation loss for decoupling scene high-frequency details and rain streaks; and
- we construct a multi-view rainy scene dataset for more realistic and consistent rain streaks.

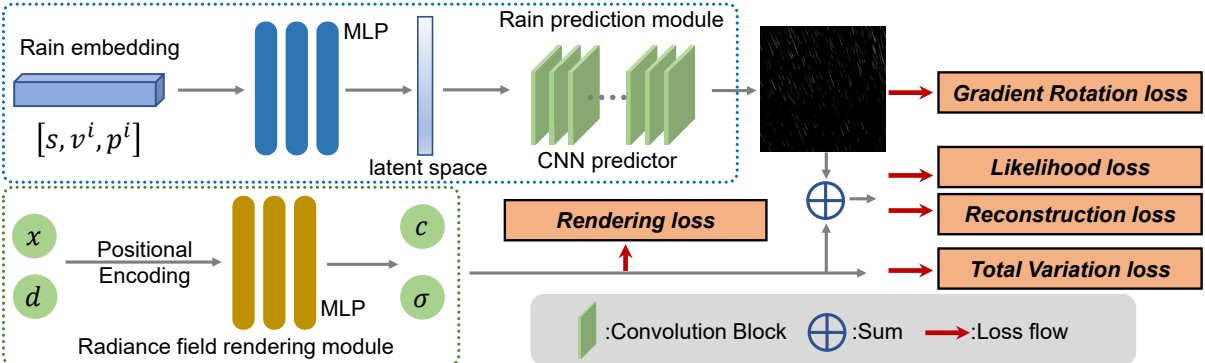

**Figure 1: Overview of the proposed RainyScape framework, which can reconstruct a rain-free scene from a set of multi-view rainy images in an unsupervised fashion. Based on the NeRF architecture, the rendering module takes ray positions and view directions as input to estimate color and density values. Rain characteristics are modeled using scene state vectors s, viewpoint state vectors $v^i$, and camera parameters $p^i$. The combined rain embedding is processed through an MLP to obtain latent space representations, which are then fed into a CNN predictor to produce a rain map. The framework is trained using unsupervised losses that facilitate the decoupling of high-frequency scene details and rain streaks, yielding a rain-free neural radiance field.**

## 2 RELATED WORK

### 2.1 Neural Rendering

NeRF [29] revolutionized viewpoint generation by introducing differentiable volume rendering and learnable radiance fields, enabling joint optimization of geometry and appearance for static 3D scenes from posed RGB images. Subsequent works explore various designs to improve NeRF's training or inference speed, such as MLP capacity [34], space discretization [10]. Recently, Kerbl et al. [19] extended the NeRF to explicit GPU-friendly 3D Gaussians Splatting (3DGS) and replaced the neural rendering, achieving real-time rendering of radiance fields. Other works focus on enhancing rendering quality and addressing challenging input scenarios. For instance, PixelNeRF [48] enables few-shot reconstruction, Mip-NeRF [2] improves rendering under multi-scale inputs, Deblur-NeRF [27] handles blurry images using canonical kernels, and NaN [31] leverages inter-view and spatial information to deal with noisy data. HDR-NeRF and RawNeRF [16, 28] utilize camera response functions and raw images to process low dynamic range inputs. In contrast, our work tackles the novel task of rainy scene reconstruction using decoupled neural rendering. We discover that the neural rendering prior is well-suited for rainy scene reconstruction and propose a general framework based on this insight, setting our approach apart from existing NeRF-based methods.

### 2.2 Image/Video Deraining

Single-image rain removal is an ill-posed problem that aims to decompose a rainy image into a clean background and a rain streak layer. Traditional methods rely on prior characteristics of rain, such as photometric properties [13], morphological component analysis [18], non-local means filtering [20], and layer priors [24] to tackle this challenge. Other approaches employ optimization techniques, including sparse coding [26], Gaussian mixture models [46], and low-rank models [8], to separate rain streaks from the background. With the advent of deep learning, numerous learning-based methods [5, 7, 11, 15, 23, 30, 32, 38, 39, 41, 49, 51] have emerged,

significantly advancing the state-of-the-art in single image rain removal performance.

Video provides additional temporal information that can be exploited for rain removal. Garg et al. [12] pioneered the task using photometric properties. Chen et al. [4] employed superpixel segmentation and a robust deep CNN for restoration. Li et al. [22] proposed a multi-scale convolutional sparse coding model to capture repetitive local patterns and rain streaks of different scales. Yang et al. introduced a two-stage recurrent framework [44] and extended it to a self-learned approach [45] leveraging temporal correlation and consistency. Yue et al [50] proposed a semi-supervised framework with a dynamical rain generator. Yan et al. developed SLDNet+ [43], an augmented self-learned deraining network utilizing temporal information and rain-related priors, and combined single-image and multi-frame modules for raindrop removal. Zhang et al. [52] proposed ESTINet, an efficient end-to-end framework based on deep residual networks and convolutional LSTMs, to capture spatial features and temporal correlations among successive frames. Xiao et al. [25] introduced a teacher-student framework for adaptive nighttime video deraining.

In addition to video-based methods, some works tackle structured multi-view image rain removal. Ding et al. [9] proposed a GAN-based architecture utilizing depth information to remove rain streaks from 3D EPIs of rainy light field images (LFIs). Yan et al. [42] employed 4D convolutions and multi-scale Gaussian processes for LFI rain removal.

### 2.3 Preliminary

In this work, we employ the classic Neural Radiance Fields (NeRF) [29] as the neural rendering method to describe our work more concisely and elegantly. NeRF is a novel view synthesis method that learns a continuous, volumetric representation of a scene through differentiable volume rendering. NeRF employs a multilayer perceptron (MLP) network $F_\Theta$ to parameterize a 5D input (3D position $\mathbf{x} = (x, y, z)$ and 2D viewing direction $\mathbf{d} = (\theta, \phi)$) into a density $\sigma$

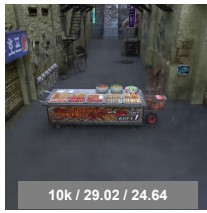 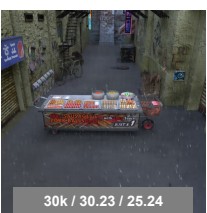 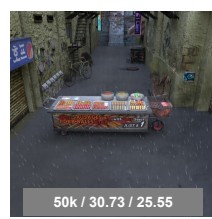 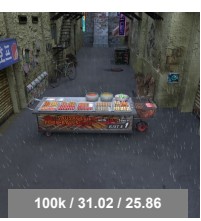 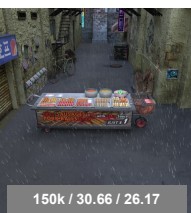 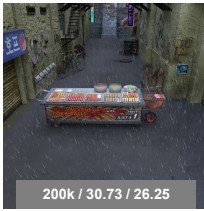

10k / 29.02 / 24.64    30k / 30.23 / 25.24    50k / 30.73 / 25.55    100k / 31.02 / 25.86    150k / 30.66 / 26.17    200k / 30.73 / 26.25

Figure 2: Illustration of neural rendering prior: From low-frequency scene representation to high-frequency detail restoration and rain preservation. The numbers represent the training epoch, the PSNR (dB) of the rendered image compared with rain-free ground truth images, and the PSNR (dB) of the rendered image compared with the rainy image.

and color $\mathbf{c} = (r, g, b)$:

$$F_\Theta : (\mathbf{x}, \mathbf{d}) \rightarrow (\sigma, \mathbf{c}) \tag{1}$$

To render a novel view, NeRF casts a ray $\mathbf{r}(t) = \mathbf{o} + t\mathbf{d}$ for each pixel, where $\mathbf{o}$ is the camera origin and $\mathbf{d}$ is the ray direction. The expected color $C(\mathbf{r})$ of the ray is computed by integrating the radiance along the ray:

$$C(\mathbf{r}) = \int_{t_n}^{t_f} T(t)\sigma(\mathbf{r}(t))\mathbf{c}(\mathbf{r}(t), \mathbf{d})dt, \tag{2}$$

where $T(t) = \exp(-\int_{t_n}^{t} \sigma(\mathbf{r}(s))ds)$ is the accumulated transmittance along the ray, and $[t_n, t_f]$ is the near and far bounds of the ray. In practice, the continuous integral is approximated using numerical quadrature:

$$\hat{C}(\mathbf{r}) = \sum_{i=1}^{N} T_i(1 - \exp(-\sigma_i\delta_i))\mathbf{c}_i, \tag{3}$$

where $T_i = \exp(-\sum_{j=1}^{i-1} \sigma_j\delta_j)$, $\delta_i = t_{i+1} - t_i$ is the distance between adjacent samples, and $N$ is the number of samples along the ray.

## 3 PROPOSED METHOD

As aforementioned, given a set of multi-view rainy images, we aim to reconstruct a rain-free scene in an unsupervised manner. As shown in Fig. 1, the proposed framework, named RainyScape, consists of a neural rendering module based on the NeRF architecture and a rain prediction module built upon MLP and CNN layers. To be specific, the neural rendering module takes ray positions and view directions as input to estimate color and density values. The rain prediction module explicitly models the rain characteristics, which consumes a learnable rain embedding composed of scene state vectors and viewpoint state vectors, as well as provided camera parameters, to predict a rain map. During training, we alternately optimize the entire framework and use unsupervised losses to encourage the decoupling of high-frequency scene details and rain streaks. This enables the reconstruction of a rain-free radiance field, which can be directly rendered during inference to obtain sharp, rain-free views. Moreover, our framework can be adapted to other rendering pipelines, such as the recent 3D Gaussian Splatting (3DGS), as experimentally demonstrated in Section 5.

**Remark**. Our RainyScape differs from existing image/video-based deraining methods in several key aspects. First, those methods rely on image space operations and do not fully exploit the scene's 3D geometry. Second, most of them require ground-truth rain-free data as supervision. Third, those methods often struggle to

handle multi-view inputs with large baselines. Furthermore, by reconstructing a rain-free radiance field, our RainyScape enables the rendering of novel views, which is not applicable to existing ones.

### 3.1 Neural Rendering Prior for Deraining

The task of deraining aims to decompose a rainy image $\mathbf{I} \in \mathbb{R}^{h \times w \times 3}$ into a rain-free background scene $\mathbf{B} \in \mathbb{R}^{h \times w \times 3}$ and a rain layer $\mathbf{R} \in \mathbb{R}^{h \times w \times 3}$ [35, 37], which can be formulated as

$$\mathbf{I} = \mathbf{B} + \mathbf{R}. \tag{4}$$

Initially, we leverage the neural rendering prior, which is fundamentally introduced by the spectral bias property of neural networks [33], to obtain a low-frequency scene representation $\mathbf{B}_l \approx \mathbf{B}$ during the warm-up stage, as illustrated in Fig. 2. Specifically, at this stage, the rainy image $\mathbf{I}$ can be decomposed into the low-frequency scene content $\mathbf{B}_l$ and the high-frequency information $\mathbf{I}_h$, as shown in the residual map in Fig. 3, which includes both rain streaks and useful scene details:

$$\mathbf{I} = \mathbf{B}_l + \mathbf{I}_h. \tag{5}$$

To obtain a more detailed background $\mathbf{B}$ from the initial low-frequency representation $\mathbf{B}_l$, we need to further decouple the high-frequency information $\mathbf{I}_h$ into rain streaks and detailed scene components. To achieve this goal, we next introduce a rain prediction module that receives and reflects the rain characteristics from the rainy scene.

### 3.2 Rain Prediction Module

To effectively model the rain characteristics and decouple the high-frequency information $\mathbf{I}_h$ into rain streaks $\mathbf{R}$ and detailed scene components $\mathbf{B}_h$, we introduce a rain prediction module $E_\Phi$. This module consists of a three-layer MLP to extract a latent space representation of the rain characteristics and a six-layer CNN to process and upsample feature maps to obtain a rain map. It takes learnable rain embeddings as input and predicts a rain map $\mathbf{R}_e \in \mathbb{R}^{h \times w \times 3}$.

The rain embeddings comprise three parts: learnable scene state vectors $\mathbf{s} \in \mathbb{R}^{128}$, learnable viewpoint state vectors $\mathbf{v} \in \mathbb{R}^{64 \times n}$, and fixed camera parameters $\mathbf{p} \in \mathbb{R}^{16 \times n}$, where $n$ is the number of input views. For a specific view $i$, the predicted rain streaks can be expressed as

$$\mathbf{R}_e^i = E_\Phi(\mathbf{s}, \mathbf{v}^i, \mathbf{p}^i). \tag{6}$$

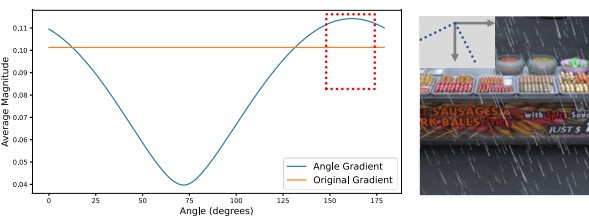

(a) Directional sensitivity of gradient magnitude differences of rainy and rain-free images.

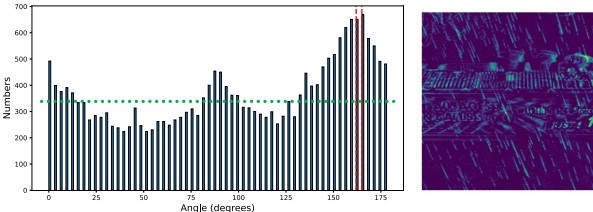

(b) Distribution of gradient orientation in the residual map.

**Figure 3: Leveraging directional sensitivity and gradient orientation for unsupervised rainy scene reconstruction. (a) Directional sensitivity of gradient magnitude differences between rainy and rain-free images. Gradients perpendicular to the rain direction exhibit higher discriminative power compared to those along the rain direction. (b) Distribution of gradient orientation in the residual map $\mathbf{I} - \mathbf{B}_l$. The residual map contains gradients in all directions (green dashed lines), with a dominant orientation perpendicular to the rain direction (red dashed lines) due to the presence of rain streaks.**

These parameters are designed to capture and represent essential information about the rain in the scene that influences the appearance of rain streaks. The rainy scene state vectors $\mathbf{s}$ represent global rainy characteristics, such as the scale and direction of the rain. The viewpoint state vectors $\mathbf{v}$ record view-specific characteristics, allowing for subtle changes in the state of rain between viewpoints. The camera parameters $\mathbf{p}$ represent the viewing direction and position, affecting the rain streaks' orientation and perspective.

## 3.3 Unsupervised Loss Function

In addition to the proposed rain prediction model, we propose a novel unsupervised loss function to help the network effectively decouple high-frequency details and rain streaks. A key component of our loss function is the adaptive gradient rotation loss, which plays a crucial role in distinguishing rain streaks from background textures.

**Adaptive Gradient Rotation Loss.** As illustrated in Fig. 3 (a), angle information can be used to differentiate rain streaks from background images. Unlike existing works, such as [17] that shifts the image through several preset angles and [49] that requires pretraining an angle estimation network, we propose an adaptive rain streak angle estimation strategy based on several experimental observations:

- The residual map $\mathbf{I} - \mathbf{B}_l$ contains high-frequency details and rain, with the angles of high-frequency details in the scene being relatively evenly distributed.

- The gradient direction of the rain is dominant after suppressing the minimum gradient.
- For local image patches, there is a high probability of rain occurring in only one direction.

Based on these observations, we calculate the orientation of the residual image gradients and construct a histogram to identify the dominant rain streak directions. Considering that the discrimination does not change significantly within a small angle range, we discretize the orientation range into 60 bins, each spanning an angle of 3 degrees for robustness. Additionally, for the special case of rain with multiple angles, we can obtain multiple angles by adjusting the number of top angles considered.

After adaptively obtaining the rain angle, for each dominant direction $\theta$, the adaptive gradient rotation loss is given by

$$\mathcal{L}_{agr} = \frac{1}{K} \sum_{j=1}^{K} (|\nabla_{\theta_k + \pi/2} \mathbf{R}| - |\nabla_{\theta_k} \mathbf{R}| + |\nabla_{\theta_k} \mathbf{B}| + |\nabla \theta_k (\mathbf{I} - \mathbf{R})|), \quad (7)$$

where $K$ is the number of dominant directions, typically set to 1, and $\nabla\theta$ denotes the gradient operator along the direction $\theta$.

**Likelihood Loss.** The likelihood loss measures the discrepancy between the reconstructed image $\mathbf{B} + \mathbf{R}$ and the input rainy image, promoting stable network performance:

$$\mathcal{L}_{ll} = \frac{1}{(\sigma^2 + \epsilon)} \cdot |\mathbf{I} - \mathbf{B} - \mathbf{R}|_2^2, \quad (8)$$

where $\sigma$ is the standard deviation of the residual, and $\epsilon$ is a small constant to prevent division by zero.

**Reconstruction Loss.** The reconstruction loss further emphasizes the consistency between the reconstructed image and the input rainy image:

$$\mathcal{L}_{rec} = |\mathbf{I} - \mathbf{B} - \mathbf{R}|_2^2. \quad (9)$$

**Total Variation Loss.** The total variation loss encourages the rendered background image to be smooth, reducing artifacts and noise:

$$\mathcal{L}_{tv} = |\nabla_x \mathbf{B}|_1 + |\nabla_y \mathbf{B}|_1, \quad (10)$$

where $\nabla_x$ and $\nabla_y$ denote the horizontal and vertical gradient operators, respectively.

The overall loss function is written as

$$\mathcal{L} = \lambda_1 \mathcal{L}_{ll} + \lambda_2 \mathcal{L}_{rec} + \lambda_3 \mathcal{L}_{tv} + \lambda_4 \mathcal{L}_{agr}, \quad (11)$$

where $\lambda_1$, $\lambda_2$, $\lambda_3$, and $\lambda_4$ are the weight coefficients for the corresponding loss terms.

## 3.4 Optimization Process

We propose an alternating optimization approach to train our framework effectively, as outlined in Algorithm 1. The optimization process consists of two main stages: network update and latent variable update. During the network update stage, we predict rain maps using the predictor $E_\Phi$, render background rays using the NeRF network $F_\Theta$, and update the network parameters $\Theta$ and $\Phi$ based on the defined loss functions Eq. (11) . In the latent variable update stage, we freeze the networks and update the latent variables $\mathbf{s}$ and $\mathbf{v}$ using Langevin Monte Carlo [14], which introduces random perturbations to explore the latent space and discover better representations for fitting the input rainy image $\mathbf{I}$. The optimization process iterates until convergence, alternating between the two stages, with

Table 1: Performance comparison of different methods on various data. The best performance is shown in bold, and the second-best performance is underlined. The *Supervised* indicates whether utilizes additional supervision information.

| Method | Supervised | Crossroad | | | Square | | | Sailboat | | | Yard | | | Average | | |
|---|---|---|---|---|---|---|---|---|---|---|---|---|---|---|---|---|
| | | PSNR ↑ | SSIM ↑ | LPIPS ↓ | PSNR ↑ | SSIM ↑ | LPIPS ↓ | PSNR ↑ | SSIM ↑ | LPIPS ↓ | PSNR ↑ | SSIM ↑ | LPIPS | PSNR ↑ | SSIM ↑ | LPIPS ↓ |
| DRSF [7]+NeRF | ✓ | 29.76 | 0.823 | 0.272 | 27.43 | 0.789 | 0.266 | 26.82 | 0.767 | 0.281 | 29.63 | 0.805 | 0.222 | 28.41 | 0.796 | 0.260 |
| EIST [52]+NeRF | ✓ | 29.54 | 0.819 | 0.256 | 27.35 | 0.789 | 0.269 | 26.42 | 0.753 | 0.307 | 29.37 | 0.796 | 0.231 | 28.17 | 0.789 | 0.266 |
| NeRF | ✗ | 27.54 | 0.836 | **0.146** | 26.44 | 0.824 | 0.136 | 26.34 | 0.822 | **0.067** | 27.39 | 0.796 | **0.099** | 26.93 | 0.820 | **0.112** |
| Ours | ✗ | **31.08** | **0.876** | 0.148 | **29.58** | **0.870** | **0.135** | **29.59** | **0.890** | 0.069 | **29.96** | **0.861** | 0.104 | **30.05** | **0.874** | 0.114 |

**Algorithm 1** Alternating Optimization for RainyScape

1: **Input:** rainy image $I$, camera poses $p$.
2: **Output:** optimized NeRF network $F_\Theta$, predictor $E_\Phi$, and latent variables $s$ and $v$.
3: Initialize $\Theta$, $\Phi$, $s$, and $v$.
4: **while** not converged **do**
5:    **if** *epoch* < 50% total epoch **then**
6:       Warm-up NeRF network $F_\Theta$.
7:    **else**
8:       **Network Update:** Predict rain maps using $E_\Phi$, Render background rays using $F_\Theta$, and Update network parameter $\Theta$ and $\Phi$ via Eq. 11.
9:       **Latent Variable Update:** Freeze $F_\Theta$ and $E_\Phi$. Update $s$, and $v$ using Langevin Monte Carlo to fit $I$ via Eq. 11.
10:    **end if**
11: **end while**

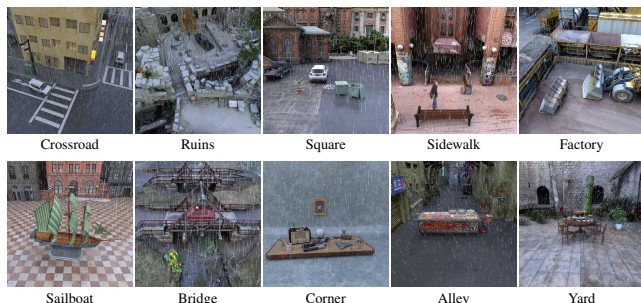

| | | | | |
|---|---|---|---|---|
| Crossroad | Ruins | Square | Sidewalk | Factory |
| Sailboat | Bridge | Corner | Alley | Yard |

Figure 4: Scenes and names of the proposed dataset.

a warm-up stage to initialize the NeRF network before introducing the rain prediction module. By leveraging this approach, our framework learns to separately represent the background scene and the rain layer, progressively refining the deraining results without relying on ground-truth supervision.

## 4 EXPERIMENTS

### 4.1 Proposed Dataset

We construct a comprehensive and diverse multi-view rainy image dataset using *Maya* [1]. As illustrated in Fig. 4, our dataset consists of 10 scenes, each captured from 50 viewpoints with $1024 \times 1024$ resolution. To ensure the view consistency of raindrops, we construct 3D models of raindrops, distribute them randomly within the scene, and render them together with the scene objects. This approach enables the generation of interaction effects between raindrops and light rays, generating more realistic rainy scenes compared to

simulation methods that directly add rain streaks to 2D images. The dataset encompasses a wide range of rain densities, directions, and streak orientations to accurately represent real-world variations. The scenes are composed of outdoor objects such as cars, bridges, roads, buildings, and boats to facilitate generalization to unseen rainy environments. The 50 viewpoints are strategically distributed on a spherical surface to maximize the overlap rate of scene content captured by different cameras, providing diverse information for rendering. The dataset includes rainy and ground truth images, depth maps, and camera parameters, making it suitable for a wide range of viewpoint synthesis tasks. *We will make the dataset publicly available.*

### 4.2 Implementation Details

Our rain prediction module consists of a 3-layer MLP with a feature dimension of 128 channels, which generates a 1024-dimensional latent space representation. This representation is then reshaped into an image and processed using a 6-layer convolutional network, followed by super-resolution to obtain the required $64 \times 64$ random patches. For the neural rendering component, we utilize the PyTorch re-implementation NeRF [47] with a batch size of 4096 rays, each ray undergoing 64 coarse samplings and 64 fine samplings. To further validate our approach, we also conduct experiments using the official implementation of 3D Gaussian Splatting (3DGS) [19] with its default configuration. During model optimization, we employ the Adam optimizer [21] with its default settings. For NeRF, we use the default learning rate update strategy, with an initial learning rate of $5 \times 10^{-4}$. After the warm-up period, the learning rate is set to $1 \times 10^{-6}$. The learning rates for the MLP and CNN part in the rain prediction module are set to $1 \times 10^{-3}$ and $1 \times 10^{-4}$, respectively. For latent variable updates, we perform 5 updates per epoch, with the first two updates using Langevin Monte Carlo to introduce random perturbations. The loss coefficients $\lambda_1$, $\lambda_2$, $\lambda_3$, and $\lambda_4$ are set to 0.1, 500, 0.5, and 1, respectively. We optimize a single model for 18K iterations on a single NVIDIA A6000 GPU.

### 4.3 Results

We set up two comparison baselines that leverage supervised information for rain removal. First, we fine-tune two state-of-the-art rain removal methods, namely the single-image rain removal method DRSformer [7] and the video-based rain removal method ESTINet [52], using six randomly selected scenes (Ruins, Sidewalk, Factory, Bridge, Corner, and Alley). These fine-tuned models are then employed as pre-processing steps for the input images. Subsequently, we train NeRF on the pre-processed images and refer to these two

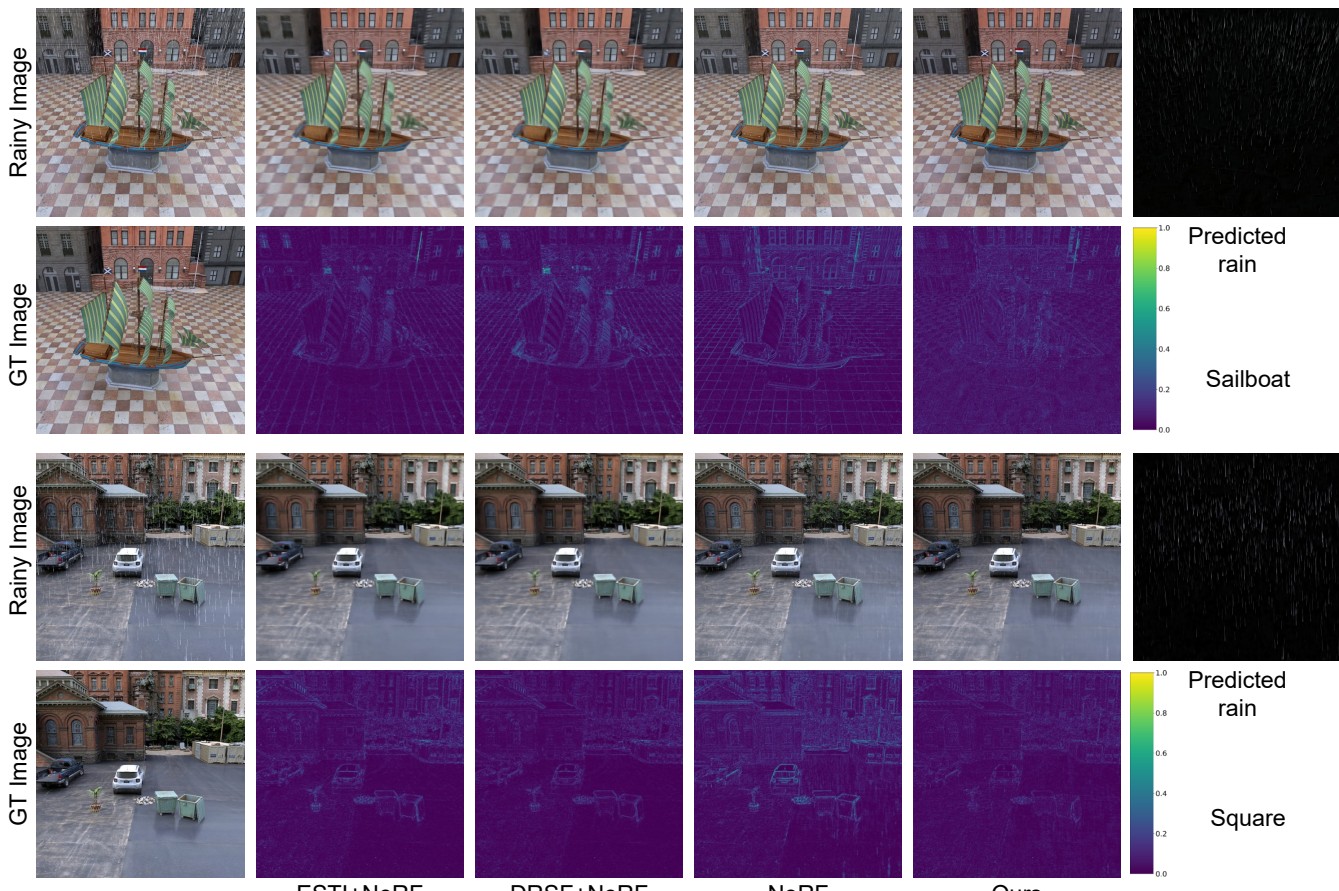

**Figure 5: Visual comparison of different methods on rainy scenes. Each scene shows rainy images, ground truth, rendered results from baseline methods and our approach, error maps, and rain streak images predicted by our method.**

approaches as DRSf+NeRF and ESTI+NeRF, respectively. Additionally, we include a baseline that directly applies the original NeRF [29] to the rainy data without any pre-processing or modifications.

We employ widely-used metrics to quantitatively evaluate the performance of our method and the baselines. Peak Signal-to-Noise Ratio (PSNR) measures the image quality and rainy scene reconstruction performance, with higher values indicating better results. Structural Similarity Index Measure (SSIM) assesses quality via structural information, luminance, and contrast, with higher values indicating better results. Perceptual similarity (LPIPS) measures the similarity in feature space, with lower values indicating better results.

**Quantitative results.** Table 1 showcases the performance comparison of our proposed method against three baselines (DRSF+NeRF, EIST+NeRF, and NeRF) on four scenes (Crossroad, Square, Sailboat, and Yard). The "Supervised" column in the table indicates whether the method utilizes supervision information. Our method achieves the best performance on the majority of the scenes and metrics (highlighted in **bold**), outperforming the baselines in most cases. On average, our approach obtains a PSNR of 29.88 dB, surpassing the second-best method, DRSF+NeRF, by 1.47 dB and the original

NeRF by 2.95 dB. We also achieve the highest average SSIM and second-best average LPIPS scores, indicating superior structural preservation and perceptual quality. It is worth noting that our method achieves state-of-the-art performance without relying on any supervision information. These quantitative results validate the effectiveness of our novel contributions and highlight its potential to render high-quality clear images across diverse rainy scenes.

**Qualitative results.** Fig. 5 presents a visual comparison of the rendering results obtained by our method and the baseline approaches. Our approach effectively renders rain-free images while accurately capturing and separating the rain streaks from the scene content. The error maps demonstrate our method's superiority in recovering high-frequency details and preserving textures and edges compared to the baselines. By explicitly modeling and decoupling the rain streaks from the high-frequency content, our framework can effectively transfer useful scene details to the radiance field, resulting in improved restoration quality. In contrast, the baseline methods often struggle to recover fine details, introducing artifacts or losing important texture information. Since the rain streaks are relatively small compared to the scene, please zoom in for better observation of the details in all figures.

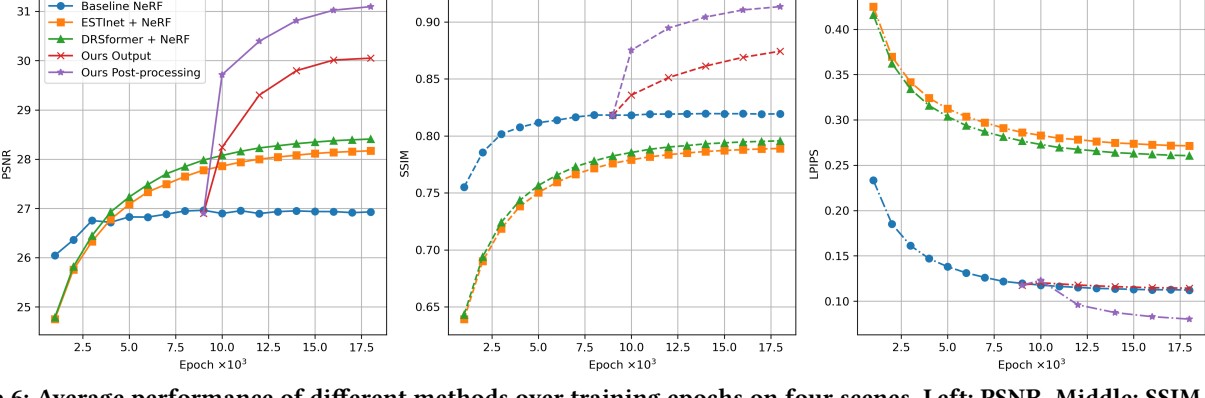

Figure 6: Average performance of different methods over training epochs on four scenes. Left: PSNR, Middle: SSIM, Right: LPIPS. Our post-processing step consistently improves the deraining quality compared to our base output.

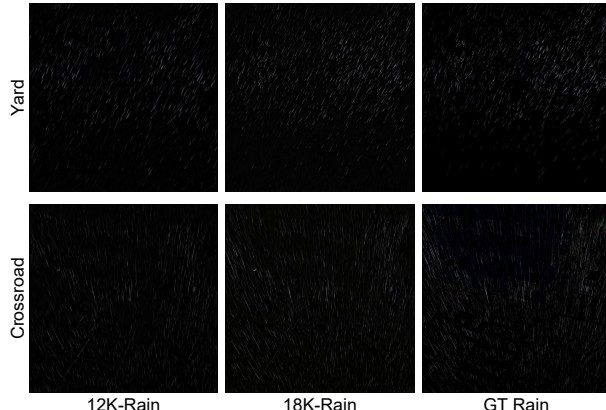

Figure 7: Evolution of predicted rain streaks during training and comparison with ground truth (GT) rain.

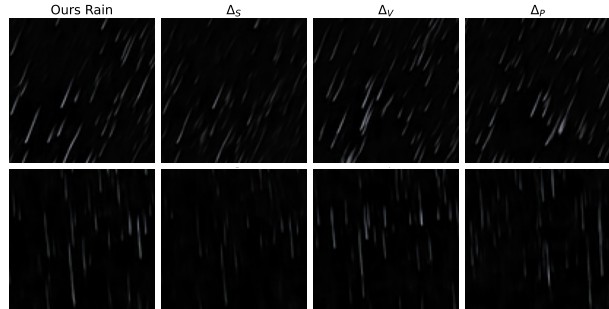

Figure 8: Impact of rain embedding changes ($\Delta_s$, $\Delta_v$ and $\Delta_p$) on predicted rain streaks, with $100 \times 100$ patch size.

**Post-processing with predicted rain.** The accuracy of the predicted rain streak images, as demonstrated in Fig. 5 and Fig. 7, enables further enhancement of the deraining performance through post-processing. When the rainy image is provided, we can apply a simple thresholding method to convert the rain streak image into a binary mask, which allows us to identify and selectively fuse regions unaffected by rain with our rendered image. This selective fusion approach leads to an improvement in the overall deraining performance. Fig. 6 illustrates the performance progression of different methods, including our post-processing step, during training. By combining the information from the rain-free regions with our RainyScape rendering results, we achieve a further performance improvement of approximately 1 dB.

## 4.4 Ablation Study

**Rain Prediction.** To demonstrate the effectiveness of our rain prediction module, we first visualize the evolution of the predicted rain streaks during training in Fig. 7. As the training progresses, the predicted rain streaks gradually resemble the real rain streaks more closely, indicating that our model effectively learns to capture the characteristics of scene rain.

Furthermore, we investigate the impact of the rain embedding on the predicted rain streaks. In Fig. 8, we present the results of

adding random Gaussian noise with a variance of 0.5 to the scene state vector **s** and viewpoint state vector **v**, denoted as $\Delta_s$ and $\Delta_v$, respectively. The results demonstrate that the added noise influences the intensity and density distribution of the predicted rain streaks, suggesting that the rain embedding plays a crucial role in controlling the appearance of the predicted rain. In addition, we show the effect of changing the camera parameters, denoted as $\Delta_p$. The predicted rain adapts to the changes in camera parameters, demonstrating the ability to predict rain streaks that are consistent with the scene information and camera setup.

**Loss configuration.** To investigate the contribution of each term in our loss function (Eq. (11)), we conduct an ablation study by removing the reconstruction loss $\mathcal{L}_{rec}$, total variation loss $\mathcal{L}_{tv}$, and adaptive gradient rotation loss $\mathcal{L}_{adg}$ terms individually. The likelihood loss $\mathcal{L}_{ll}$ is kept in all configurations to ensure stable training. Additionally, we explore the impact of varying the number of bins used in the adaptive gradient rotation loss, considering 30, 60 (default), and 90 bins. All the results are shown in Table 2. Notably, when the $\mathcal{L}_{adg}$ is not used, the network fails to predict visually realistic rain streaks, emphasizing its critical role in capturing the directional characteristics of rain.

**Framework Effectiveness.** We compare our framework with two alternative approaches to demonstrate its effectiveness. The first approach, Two-NeRF, uses separate neural rendering fields for rain and background, similar to Dehaze-NeRF [6]. However, as shown in Fig. 10, Two-NeRF fails to effectively decouple rain

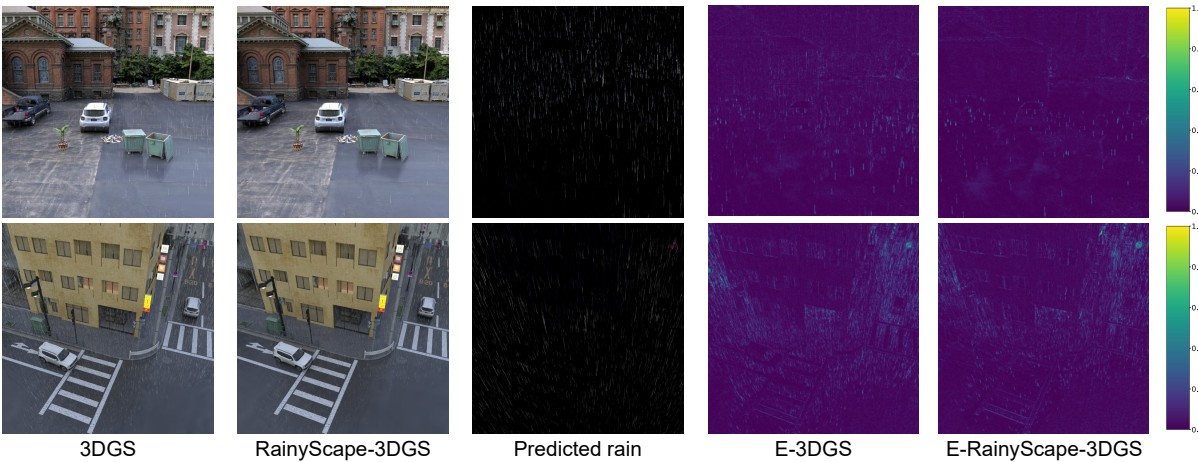

| 3DGS | RainyScape-3DGS | Predicted rain | E-3DGS | E-RainyScape-3DGS |

**Figure 9: Visual results of the extension of our RainyScape to 3DGS. The error maps are denoted as E-∗.**

**Table 2: Ablation study of loss configuration on the Yard data.**

| Method | PSNR ↑ | SSIM ↑ | LPIPS ↓ |
|---|---|---|---|
| w/o $\mathcal{L}_{rec}$ | 12.52 | 0.428 | 0.467 |
| w/o $\mathcal{L}_{tv}$ | 29.51 | 0.850 | 0.108 |
| w/o $\mathcal{L}_{agr}$ | 29.28 | 0.856 | 0.105 |
| 30 bins | 29.63 | 0.859 | 0.106 |
| 90 bins | 29.60 | 0.858 | 0.106 |
| Ours | 29.96 | 0.861 | 0.104 |

**Table 3: Average performances of 3DGS extension.**

| Method | PSNR ↑ | SSIM ↑ | LPIPS ↓ |
|---|---|---|---|
| 3DGS | 32.41 | 0.907 | 0.219 |
| RainyScape-3DGS | 35.66 | 0.947 | 0.067 |
| RainyScape-3DGS (Post-Process) | 36.48 | 0.952 | 0.056 |

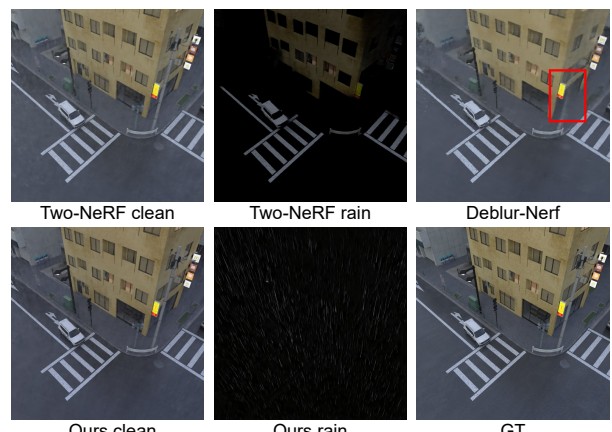

| Two-NeRF clean | Two-NeRF rain | Deblur-Nerf |
| Ours clean | Ours rain | GT |

**Figure 10: Limitations of alternative approaches. Two-NeRF fails to decouple rain and background. Deblur-NeRF damages scene structure (red box).**

from the background, while our method accurately distinguishes between the two components. We also evaluate Deblur-NeRF [27], which adjusts and fuses rays to handle motion blur. As highlighted in Fig. 10, Deblur-NeRF severely damages the scene structure, erroneously removing objects like utility poles.

## 5 EXTENSION TO 3D GAUSSIAN SPLATTING

3D Gaussian splatting (3DGS) [19] has emerged as a promising technique in the field of radiance field rendering, offering high efficiency and impressive visual quality. In this section, we explore the extension of our RainyScape framework to incorporate 3DGS as the rendering module. By leveraging the capabilities of 3DGS, we aim to enhance the deraining performance while maintaining the computational efficiency and visual fidelity of the rendered results.

To extend our framework to 3DGS, we replace the NeRF rendering component in the pipeline with the 3DGS architecture, getting RainyScape-3DGS, while the rain prediction modules remain unchanged. To ensure optimal performance, the warm-up stage is set to 4K iterations. Subsequently, joint training of the entire framework is performed using the default configurations and loss functions. We evaluate the effectiveness of RainyScape-3DGS using the same datasets. Table 3 presents the quantitative results, proving that our RainyScape framework can significantly improve the performance of 3DGS in rainy scenes. Fig. 9 provides visual comparisons, showcasing the ability of our framework to suppress the rendered rain in 3DGS and more visually appealing rain-free images.

## 6 CONCLUSION

In this paper, we introduced RainyScape, a novel unsupervised framework for reconstructing scenes from multi-view rainy images using decoupled neural rendering. Our approach tackles the challenges of rain streak removal in the rendering process by integrating a low-frequency scene representation, a rain prediction module with a learnable rain embedding, and an unsupervised rainy scene reconstruction loss that incorporates an adaptive gradient rotation loss. Extensive experiments on both classic NeRF and state-of-the-art 3DGS demonstrate the effectiveness and versatility of RainyScape in generating clean, visually appealing images with sharp details and accurate scene geometry.

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
