# OpenReview forum: "RainyScape: Unsupervised Rainy Scene Reconstruction using Decoupled Neural Rendering"
_acmmm.org/ACMMM/2024/Conference — MM2024 Poster_

### Official Review · Reviewer_mn4k · 2024-05-16

**Rating:** 4
**Confidence:** 3

**Summary:**

This work proposes to reconstruct rainy scenes under Neural Rendering in an unsupervised spirit. They design a framework RainyScape, which introduces rain embedding to predict rain streaks and an adaptive gradient rotation loss to decouple high-frequency details and rain streaks. A new multi-view rainy scene dataset is also constructed for better benchmarking.

**Strengths:**

It is interesting to investigate rainy scene reconstruction under NeRF and 3DGS.
The dataset is a great contribution to the community of image deraining.
It is practically useful to develop an unsupervised method for image deraining, which does not need paired data.
Superior results are achieved in diverse scenarios.

**Limitations:**

1. The contribution (ii) should be described in detail. Its motivation and novelty are unclear. Why do we need rain embedding? Could you provide more description on this point?
2. The proposed gradient rotation loss, the author states the characteristic of 'adaptive', differing from existing work. More details and motivation should be given to explain 'adaptive'.
3. More details and analysis of Fig. 5 are needed. It is hard to observe that the proposed method is superior in recovering high-frequency details and preserving textures and edges from the displayed error maps. How the error maps come from?
4. Analysis on the sensitivity of balancing hyper-parameters is missing. It would be great to investigate how the proposed Adaptive Gradient Rotation Loss contributes to the results.

**Suitability:**

3

---

### Official Review · Reviewer_ZnnR · 2024-05-20

**Rating:** 5
**Confidence:** 4

**Summary:**

The paper introduces the RainyScape framework, which combines neural rendering and rain-prediction modules to achieve unsupervised clear scene reconstruction from multi-view rainy images. It utilizes neural rendering priors to obtain a low-frequency scene representation by optimizing the neural rendering pipeline, effectively removing rain streaks and generating clear images.
The paper introduces an adaptive gradient-based reconstruction loss to drive the joint optimization of the two modules.
The paper constructs a multi-view rainy scene dataset for more realistic and consistent rain streaks.
Experimental results demonstrate the efficacy of the proposed method.

**Strengths:**

1. The authors generate a diverse and high-quality multi-view video derain dataset for training and evaluation purposes in the context of rainy scene reconstruction.
2. The Adaptive Gradient Rotation Loss enhances the network's ability to effectively decouple high-frequency details and rain streaks. The proposed strategy in the paper is adaptive and based on experimental observations.

**Limitations:**

1. The reviewer recommend the authors to provide more detailed explanations about the operations of rain embedding to enhance the understanding of readers.
2. There are some typos in this article. For example, there is an extra 'and' on line 111. Please further refine.
3. It is recommended to include additional references from recent works in video deraining to enhance the comprehensiveness of paper. Here are some suggested references:
(1) Sun S, et al. Event-aware video deraining via multi-patch progressive learning[J]. IEEE Transactions on Image Processing, 2023.
(2) Wang J, et al. Unsupervised Video Deraining with An Event Camera[C]//Proceedings of the IEEE/CVF International Conference on Computer Vision. 2023.
(3) Wu H, et al. Mask-Guided Progressive Network for Joint Raindrop and Rain Streak Removal in Videos[C]//Proceedings of the 31st ACM International Conference on Multimedia. 2023.
4. Additionally, it is better to compare your work with more representative video deraining methods, such as SLDNet, and other NeRF-based video restoration methods, if available.

**Suitability:**

3

---

### Official Review · Reviewer_pyHj · 2024-05-24

**Rating:** 3
**Confidence:** 3

**Summary:**

This work introduce an adaptive scene rain streak angle estimate strategy and a corresponding gradient rotation loss for rainy scene reconstruction. In addition, it provides a datasets.

**Strengths:**

The visualization results seems good, and the proposed Adaptive Gradient Rotation Loss is reasonable and is detailed.

**Limitations:**

1. What is the application environment of rainy scene reconstruction task? In the introduction, you only show the importance of your task by listing previous other tasks. I am interested in understanding the real-world scenarios where this task would provide substantial benefits or improvements over existing solutions.

2. I do not have a clear understanding about how to get the low-frequency scene content B_l and the high-frequency information I_h. Does this work manually choose a image in one epoch to obtain them? What exact the epoch? Is the epoch selected for each scene the same? Can it be selected automatically?

3. Adaptive Gradient Rotation Loss is proposed based on the assumption that the gradient direction of the rain is dominant after suppressing the minimum gradient. If the rain line is smaller than it in the image of the article, will the gradient direction still be dominated by rain? Does this method still work?

4. Is there any comparison with the single-image rain removal method alone? In Figure 5, you provide the result combining single-image rain removal method and Nerf. I do not know whether the image is novel view or input view. Therefore, I want to see the compare result of input view, if we only use rain removal methods.

5. Since one of the contribution is that representing the rain in the scene using rain embeddings, can this work provide quantitative ablation study to learnable scene state vector s, learnable viewpoint state vectors v , and fixed camera parameters p, separately?

**Suitability:**

2

---

### Official Review · Reviewer_bkg5 · 2024-05-25

**Rating:** 2
**Confidence:** 3

**Summary:**

This paper introduced an unsupervised framework for reconstructing clean scenes from multi-view rainy images. RainyScape includes a neural rendering module and a rain-prediction module with a predictor network and a learnable latent embedding for rain characteristics. By leveraging the spectral bias property of neural networks, the neural rendering  is optimized to obtain a low-frequency scene representation. Extensive experiments show that it eliminated rain streaks and rendering clean images.

**Strengths:**

It provided a clear picture to show the proposed RainyScape framework.
It provided diverse applications.

**Limitations:**

Sections 3.1 and 3.2 have little novelty, with the main contributions coming from the unsupervised loss functions. However, the paper lacks clear motivations, significantly reducing its overall quality.

Why do the PSNR-SSIM-LPIPS results of the proposed method only start to show improvement around epoch 9000?

PSNR, SSIM, and LPIPS are typically used to evaluate image noise quality. Are they appropriate metrics for measuring deraining performance?

It is difficult to discern details in Figures 7 and 8. What is the specific purpose of Figure 7?

Many deraining papers are published annually, and this paper lacks compelling reasons for acceptance. Additionally, it is only marginally related to MM, and  suitable for CV.

**Suitability:**

2

---

### Meta-Review · Area_Chair_rD77 · 2024-07-02

**Recommendation:** Accept (Poster)
**Confidence:** 4

**Metareview:**

This paper presented a method for rainy scene reconstruction from multi-view images. A neural rendering module and a rain prediction module were proposed to achieve the goal. A multi-view rainy dataset was also contributed. The paper received comments from four reviewers. The strengths of the paper include: the idea of rainy scene reconstruction is interesting, the contributed dataset, and the good performance. Limitations include:  unclear descriptions for some claims/statements, potential applications, insufficient comparison with other related methods, and some writing issues. The paper received mixed recommendations (both negative and positive) before the rebuttal, but after the rebuttal, most concerns were well addressed and the recommendation scores were increased.

Considering the above, the AC would recommend Accept, but suggests the authors carefully revise their paper according to the comments from all the reviewers.